# Quality of Life and Bowel Function Following Early Closure of a Temporary Ileostomy in Patients with Rectal Cancer: A Report from a Single-Center Randomized Controlled Trial

**DOI:** 10.3390/jcm10040768

**Published:** 2021-02-15

**Authors:** Audrius Dulskas, Vidas Petrauskas, Justas Kuliavas, Klaudija Bickaite, Mikalojus Kairys, Kastytis Pauza, Alfredas Kilius, Egidijus Sangaila, Rimantas Bausys, Eugenijus Stratilatovas

**Affiliations:** 1Departament of General and Abdominal Surgery and Oncology, National Cancer Institute, LT-08406 Vilnius, Lithuania; justas.kuliavas@gmail.com (J.K.); kastytis.pauza@nvi.lt (K.P.); Alfredas.kilius@nvi.lt (A.K.); egidijus.sangaila@nvi.lt (E.S.); rimantas.bausys@nvi.lt (R.B.); eugenijus.stratilatovas@nvi.lt (E.S.); 2Faculty of Medicine, Vilnius University, LT-03101 Vilnius, Lithuania; vidas.petrauskas@mf.stud.vu.lt (V.P.); claudia.bickaite@gmail.com (K.B.); mikalojuskairys@gmail.com (M.K.)

**Keywords:** early closure, ileostomy, rectal cancer, quality of life, LARS

## Abstract

The aim of this study was to assess quality of life and bowel function in patients undergoing early vs. standard ileostomy closure. We retrospectively assessed patients from our previous randomized controlled trial. Patients with a temporary ileostomy who underwent rectal cancer surgery and did not have anastomotic leakage or other. Early closure (EC; 30 days after creation) and standard closure (SC; 90 days after creation) of ileostomy were compared. Thirty-six months (17–97) after stoma closure, we contacted patients by phone and filled in two questionnaires—The European Organization for Research and Treatment of Cancer Quality of Life Questionnaire Core 30 (EORTC QLQ-C30) and low anterior resection syndrome (LARS) score. This index trial was not powered to assess the difference in bowel function between the two groups. All the patients in the SC group had anastomosis <6 cm from the anal verge compared to 42 of 43 (97.7%) in the EC group. There were no statistically significant differences between EC (26 patients) and SC (25 patients) groups in the EORTC QLQ-C30 and LARS questionnaires. Global quality of life was 37.2 (0–91.7; ±24.9) in the EC group vs. 34.3 (0–100; ±16.2) in the SC (*p* = 0.630). Low anterior resection syndrome was present in 46% of patients in the EC and 56% in the SC group (*p* = 0.858). Major LARS was found more often in younger patients. However, no statistical significance was found (*p* = 0.364). The same was found with quality of life (*p* = 0.219). Age, gender, ileostomy closure timing, neoadjuvant treatment, complications had no effect of worse bowel function or quality of life. There was no difference in quality of life or bowel function in the late postoperative period after the early vs. late closure of ileostomy based on two questionnaires and small sample size. None of our assessed risk factors had a negative effect on bowel function o quality of life.

## 1. Introduction

Nowadays advanced, but still potentially curable rectal cancer of the middle and lower thirds are treated with low anterior resection with total mesorectal excision [1]. The most feared complication—anastomotic leakage—is associated with increased postoperative mortality and worse oncological outcomes [2,3]. A diverting stoma is sometimes recommended after low anterior resection because it reduces the rate and severity of complications, especially anastomotic leakage [4,5]. Typical stoma closure time is at least three months following the resection [6,7], and it may have a negative impact on additional morbidity and impair quality of life, psychological and emotional state of the patient [8,9]. This, in turn, can influence treatment compliance and completeness [10]. Early stoma closure within two weeks after resection has been advocated to be safe in selected patients in randomized trials [11,12]. Many patients following surgery for rectal cancer develop low anterior resection syndrome (LARS) [13,14]. It is thought that neural damage, fibrosis loss of rectal reservoir and altered colonic motility are the main reasons for the development of bowel dysfunction [15]. On the other hand, diverting stoma is also associated with LARS because of bowel inflammation caused by altered colonic nutrition, changes in bacterial flora and atrophy of neural terminals in the bowel wall [16]. Theoretically, early closure of stoma should prevent this bowel damage. Still, there is a lack of strong data regarding ileostomy closure time’s effect on quality of life and bowel function.

Previously we have published a study assessing the early outcome of the early closure (EC) group vs. standard closure (SC) [17]. The study was terminated because the overall 30 days postoperative morbidity rate was dramatically higher in the EC group (27.9% vs. 7.9%; *p* = 0.024). Moreover, severe complications (Clavien–Dindo ≥ 3) were present only after the EC of ileostomy in five (11.6%) patients.

An anastomotic leak after low anterior resection with total mesorectal excision for rectal cancer causes morbidity and mortality up to 6–22% of patients and can occur in patients with no obvious risk factors [2,18]. Forming a diverting stoma does not prevent an anastomotic leak. However, it does avert the potential morbidity and mortality from an anastomotic leak [19]. On the other hand, there are many complications related to diverting stoma. Some of these include small bowel obstruction [20], acute and chronic kidney injury due to high stoma output, a parastomal hernia or complications of stoma reversal operation [21]. This is why some surgeons close the ileostomy in less than two weeks. In selected patients, the results are promising, and some authors suggest that the early closure is safe. In a prospective non-randomized study with 36 patients included, Menegaux et al. found no difference in complications 10 days closure vs. delayed (>2 months) closure group [22]. Alves et al., in a randomized clinical trial with 186 patients included, reported similar morbidity rates over 3 months between 8 days and 2 months ileostomy closure groups (respectively 31 vs. 38%) [12]. Danielsen et al. in a multicenter randomized controlled trial with 112 patients, showed that the mean number of complications following 12 months from ileostomy closure in the early closure group (8–13 days after stoma creation) is even lower than in the late closure (>3 months) group (1.2 vs. 2.9, respectively, *p* < 0.0001) [11]. On the other hand, our previously published study showed opposite results: morbidity rate was much higher in the early closure (1 month) group compared with the late closure (3 months) group (27.9% vs. 7.9%, respectively) [17]. The explanation for this may be the timing of the early closure, which was one month. Most authors would argue this being the worst time for stoma closure, and we definitely agree with that. Obviously, the question of perfect ileostomy closure time is still open.

The aim of this study was to conduct a secondary analysis of our previously published study [17] using two questionnaires—LARS score and The European Organization for Research and Treatment of Cancer Quality of Life Questionnaire Core 30 (EORTC QLQ-C30)—to investigate if early (1 month) vs. late (3 months) stoma closure has a different impact on bowel function and quality of life in the late postoperative period. Moreover, to assess whether a higher complication rate affects the quality of life (QoL) or bowel function.

## 2. Materials and Methods

Vilnius Regional Bioethics Committee gave the approval for the study. Trial number in clinicaltrials.gov—NCT03796702. All the patients signed the consent for participation in the study.

During the study period from December 2011 to December 2017, patients who underwent rectal resection with a temporary ileostomy for cancer at National Cancer Institute (Vilnius, Lithuania) were included. All the patients with middle or lower rectal cancer underwent low anterior resection with total mesorectal excision and anastomosis lower than 6 cm. If there was a locally advanced tumor, preoperative chemoradiotherapy was initiated. Patients were randomized using a computer-generated block of six to two groups: early closure group (30 days following the surgery) vs. late closure group. On the 7th–10th postoperative day, contrast enema or/and endoscopy and digital exam were performed for a possible leak. If the leak was suspected—patients were excluded from the study. Ileostomy closure was performed according to our department guidelines: all the anastomoses were performed in a single layer non-interrupted suture. The skin was closed with a purse-string suture. None of the patients in the early closure group received adjuvant chemotherapy before the closure. In the standard closure group, if adjuvant chemotherapy was needed, patients underwent closure between the cycles (one week following one of the cycles of chemotherapy).

We contacted the patients after the closure of diverting stoma (median 36 months, range 17–97 months) by phone and filled in two questionnaires: EORTC QLQ-C30 (version 3.0) [23] and LARS score [24]. Lithuanian version of the low anterior resection syndrome questionnaire was used [25]. The data manager assessing quality of life and bowel function were blinded.

If the patient had verified leak, general contraindications for the surgery, he was excluded from the study.

Questionnaires were not completed if it was not possible to contact the patient by phone, if the patient was dead or if the patient had a permanent stoma (EC group—17 patients, SC—13 patients) (Figure 1).

This study’s primary outcome was the comparison of quality of life and prevalence of LARS between the groups.

In our previous study, 86 patients (43 from the EC and 43 from the SC groups) were enrolled. Seventy-seven were excluded: 37 for the suspected leak, 18 declined to participate, 22 for other reasons (including change in the surgical procedure: no ileostomy formed, Hartmann’s procedure performed, organizational difficulties, ileostomy not closed). In this study, EORTC QLQ-C30 and LARS questionnaires were filled by 26 patients from the EC and 25 patients from the SC group. Overall data from 35 (40.7%) patients (17 from the EC and 18 from the SC group) was lost because of death, or they could not be contacted by phone nor had a permanent stoma (Figure 1). We have also assessed if complications, age, the timing of ileostomy closure, neoadjuvant treatment, gender were risk factors for the worse long-term quality of life and bowel function. In addition, we analyzed if major LARS caused the worse quality of life.

### Statistical Analysis

The sample size was calculated using G*Power 3.1.9.4 sample size calculator free version available from https://stats.idre.ucla.edu/other/gpower/ (accessed on 10 December 2020). The value of alpha—the probability of a false positive was set at 5% and hence the familiar *p* < 0.05. Power is 1-beta, so in percentage terms, these were expressed as 80%. The effect size was set at 0.15 (the expected difference of patients having major LARS between the two groups of 15%). For 1:1 randomization, it showed that 44 patients (22 in each arm) would provide 80% power for a two-sample proportion test. There are likely to be patients lost to follow-up, so the target recruitment was set at 50.

We used an intention-to-treat principle for the data analysis. For the Gaussian quantitative variable, Student’s *t*-test was used. For the non-Gaussian variable, we used the Mann–Whitney U test. *p* < 0.05 were considered statistically significant. For risk factors, we used ANOVA (univariate) and MANOVA (multivariate) statistical analyses for continuous variables. For categorical variables, a chi-squared test was used. All the statistical analysis was performed using IBM SPSS Statistics for Windows, Version 23.0 (IBM Corp: Armonk, NY, USA).

## 3. Results

The demographics of our study can be found in Table 1.

EORTC QLQ-C30 questionnaire showed no statistically significant difference between the early and late closure group in any of the longitudinal scales (Figure 2). Most of the patients evaluated their physical and cognitive functioning positively—80% and 85.8% in the EC vs. 79.7% and 84.6% in the SC group. Comparing all functional scales, social functioning was lowest—69% in both groups. The patients responded that the main symptoms they had were insomnia (34.6% in the EC and 45.8% in the SC group), diarrhea (38.4% in the EC and 29.3% in the SC group) and fatigue (33.1% in the EC and 34.6% in the SC group). The mean global quality of life in the EC group was 37.2% and 34.3% in the SC group.

Low anterior resection syndrome was present in 46% of patients in the EC and 56% in the SC group (Table 2). Major LARS manifested in 14 patients. Six of them (23%)—in the EC group, and eight (32%)—in the SC group. There was no statistically significant difference between the groups (*p* = 0.858).

Major LARS was found more often in younger patients. However, no statistical significance was found (*p* = 0.36). The same was found with quality of life (*p* = 0.22).

Major LARS had a significant negative effect in most areas of the EORTC QLQ-C30 questionnaire (Figure 3).

None of the assessed risk factors had a negative effect on bowel function or quality of life (Table 3).

## 4. Discussion

The results of our study showed that 49% of the patients following stoma closure in the long postoperative period had some degree of bowel dysfunction: 22% of them had minor and 27%—major LARS. Age, the timing of ileostomy close, complications, neoadjuvant treatment, and gender had no negative effect on bowel function or quality of life. The same results are reported in other controlled clinical trials. Keane et al., in a multicenter RCT with 112 participants, investigated late postoperative (median 49 months) bowel function after the early (8–13 days) and late (>3 months) stoma closure. The patient drop was also high, reaching almost 30%. Major LARS rates were higher in the late closure group but did not reach statistical significance (72% vs. 59%, respectively, *p* = 0.25) [26]. In our study, major LARS rates were lower—30% in the late closure and 24% in the early closure group.

EORTC QLQ-C30 questionnaire showed similar results as in the literature regarding functional and symptom scales [27]. Herrle et al. evaluated quality of life six months after temporary diverting stoma closure in an observational, multicenter study of 120 patients using the EORTC QLQ-C30 questionnaire [27]. The median time to stoma closure was 5 months (range, 17 days—18 months), and 3.4% of patients had very early stoma closure within 30 days. Functional scales were comparable to our study. However, our patients responded that their global quality of life was 36% (worse) compared with 60% reported by Herrle et al. [27]. In our study, quality of life might be low in both EC and SC groups not because of ileostomy closure time. It can be associated with general low self-esteem. Obviously, it would have added more scientific value assessing quality of life in few earlier time spots (for example, just after the rectal resection, 3–6 months later, and after the ileostomy takedown)—as the caring the ileostomy and some ileostomy complications (as watery diarrhea, dehydration, or electrolyte abnormality) definitely affect the living. None of our patients had any of these ileostomy related complications. Later quality of life should be very similar and can be affected by bowel function. A clinical trial by Scarpa et al. suggests that the male gender is associated with the higher evaluation of quality of life with an ileostomy [28]. In our study, males accounted for 56.6%, but the general quality of life was still low. We found no difference in the early and late closure of diverting ileostomy groups in any scale of the EORTC QLQ-C30 questionnaire. In the literature, no relevant associations could be found between time to stoma closure and quality of life in a long perspective [27,29,30]. EASY trial by Park et al. showed promising results in the early postoperative period after the early stoma closure. However, quality of life using three different questionnaires (EORTC QLQ-C30, EORTC QLQ-CR29 and SF-36) filled 3, 6, and 12 months after resection showed no difference between early and late closure groups [30]. Zhen et al. found no difference in quality of life between early (3 months) and late (6 months) closure groups, but suggested that the late closure is preferable if adjuvant chemotherapy is indicated because it favors the implementation of standard therapy, which is recommended for locally advanced tumors [29]. We have not assessed the possible risk factors (height of anastomosis, preoperative radiotherapy) for worse bowel function or QoL as the sample size is quite small, and this was not the aim of this study. Nevertheless, both groups were statistically the same comparing height of anastomosis and preoperative treatment.

In a similar study by Jiménez-Rodríguez et al., the patient drop was even higher than in our index trial (>50%, compared to our 47%) [31]. The authors assessed the risk factors for worse bowel function following the ileostomy takedown. They showed that the interval to ileostomy closure was not a risk factor for LARS. In multivariate analysis, male gender and preoperative neoadjuvant therapy were significant predisposing factors for LARS.

The main strength of our study is a randomized design. Second, we have assessed the bowel function using the new LARS score with the EORTC quality of life questionnaire. Moreover, we should that although the rate of complications was much higher in the early closure group, it does not translate into worse bowel function or quality of life.

However, it is a secondary analysis of our previous randomized trial. Two main limitations of our study are the small sample size (51) and the number of lost patients (35–40.7%) what increases the risk of bias and makes analysis underpowered to answer the bowel function question. Another limitation is the long inclusion period and wide range of time to follow-up visits (up to 97 months). In addition, a longitudinal assessment would have been preferable over cross-sectional analysis. This is obvious with quality of life, which is negatively affected, while period caring the ileostomy. The different scenario is with the bowel function, as our and other authors’ results showed that LARS is a long-term issue, without any significant change in a timeline [32,33,34,35,36,37]. In addition, we did not evaluate sexual and urinary functions that are also important parts of quality of life.

## 5. Conclusions

We found no difference in quality of life or bowel functioning in the late postoperative period after the early vs. late closure of ileostomy based on two questionnaires and small sample size. None of our assessed risk factors had a negative effect on bowel function o quality of life. A very large RCT with a non-inferiority design must be performed assessing quality of life, and bowel function as a primary outcome should be designed.

## Figures and Tables

**Figure 1 jcm-10-00768-f001:**
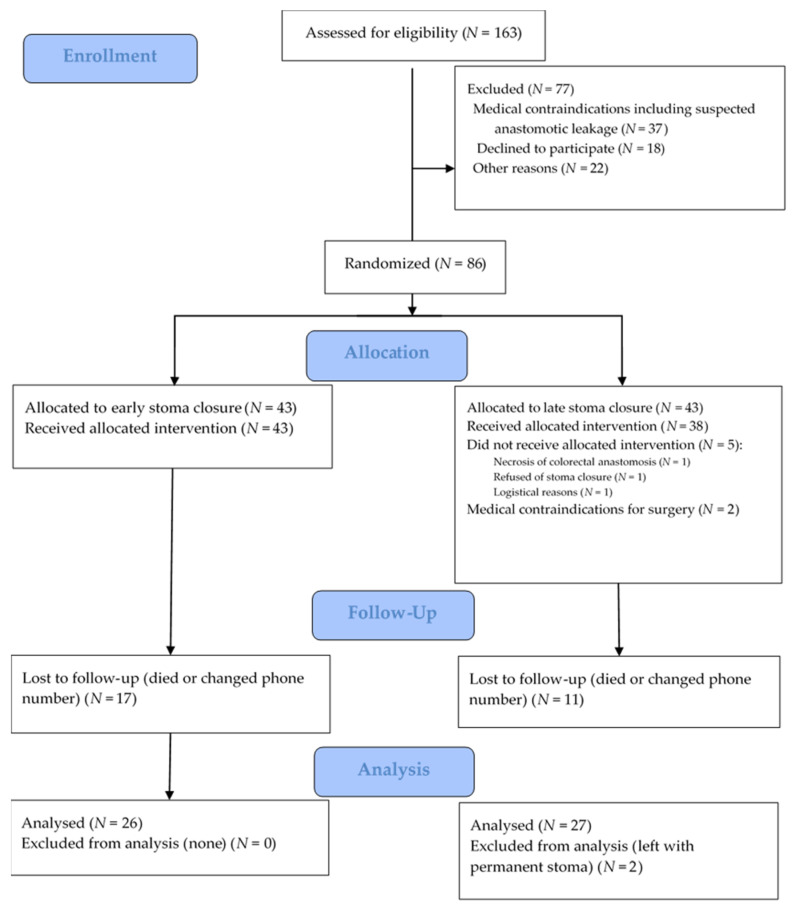
CONSORT flow diagram of patients included in the study.

**Figure 2 jcm-10-00768-f002:**
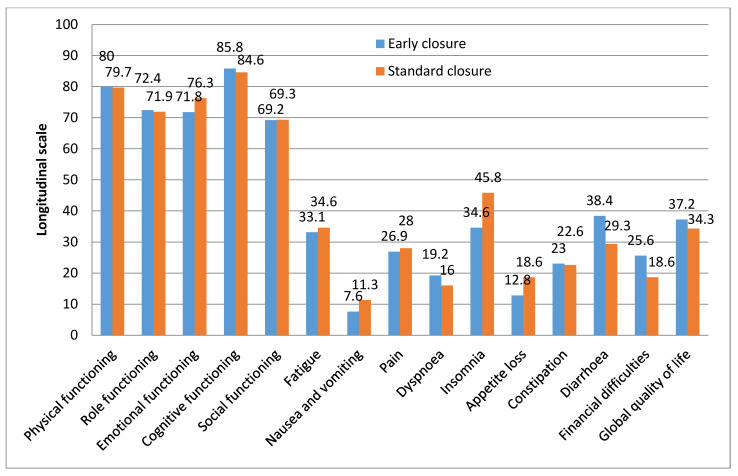
European Organization for Research and Treatment of Cancer Quality of Life Questionnaire Core 30 (EORTC QLQ-C30) longitudinal function, symptom and global health scales for patients undergoing early or standard ileostomy closure *. * Data were obtained from the two groups of patients: early closure group *N* = 26, standard closure group *N* = 25, using a validated questionnaire EORTC QLQ-C30. QLQ-C30 is composed of 30 items assessing global perceived health status and QoL. These items are grouped in five functional scales—physical functioning (PF), role functioning (RF), emotional functioning (EF), cognitive functioning (CF) and social functioning (SF); three symptom scales—fatigue (FA), nausea and vomiting (NV) and pain (PA); six single-item scales—dyspnea (DY), insomnia (SL), appetite loss (AP), constipation (CO), diarrhea (DI) and financial difficulties (FD). Scales are converted to a score ranging from 0 to 100. The higher the scores of the overall QoL and functioning scales indicate the better the overall QoL and functioning; however, the higher the scores of the symptom scales indicate the lower QoL.

**Figure 3 jcm-10-00768-f003:**
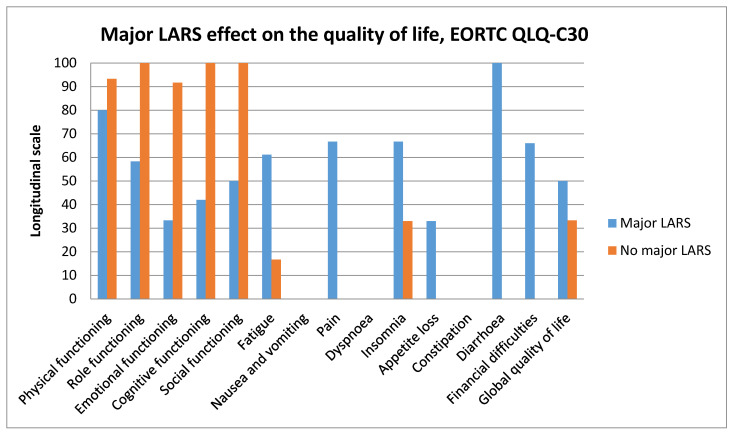
Major low anterior resection syndrome (LARS) effect on quality of life *. * Two groups: 26 patients in the early closure group and 25 in the standard closure group compared. For quality of life assessment, EORTC QLQ-C30 was used. For bowel function (low anterior resection syndrome—LARS), assessment validated low anterior resection syndrome score was used. It consists of five questions related to bowel function following the low anterior resection: incontinence for flatus and/or liquid stool, frequency of bowel movements, clustering of the stools, and fecal urgency. The score of <21 points means no LARS, score from 21 to 29 points—minor LARS, whereas 30 to 42 points indicate major LARS. Independent samples Mann–Whitney U test was used.

**Table 1 jcm-10-00768-t001:** Baseline demographic and clinicopathologic characteristics of the study groups *.

Variable	Early Closure Group (*N* = 26)	Standard Closure Group (*N* = 25)	*p* Value
Timing of ileostomy closure, days (range), median	34 ± 15 (from 29 to 47), 38	92 ± 25 (from 80 to 157), 90	0.001
Timing from ileostomy closure to questionnaires filling (months), median	38 ± 16 (from 17 to 97), 30	37 ± 15 (from 17 to 86), 32	0.87
Sex, male/female (N, %)	14 (53.8%)/12 (46.2%)	11 (44%)/14 (56%)	0.62
Age, years (range), median	63 ± 9.4 (from 56 to 68), 60	65 ± 9.3 (from 60 to 68), 63	0.81
Comorbidity (N, %)	20 (76.9%)	19 (76%)	0.9
Cardiac diseases	10 (50%)	11 (57.9%)	
Diabetes	2 (10%)	2 (10.5%)	
Pulmonary diseases	1 (5%)	1 (5.3%)	
Charlson comorbidity Index (range), median	4 ± 1.5 (from 4 to 6), 4.5	4 ± 1.25 (from 3.75 to 6), 4.5	1
Stage of disease (N, %)			0.78
I	10 (38.5%)	9 (36%)	
II	10 (38.5%)	7 (28%)	
III	5 (19.2%)	8 (32%)	
IV	1 (3.8%)	1 (4%)	
Neoadjuvant chemoradiotherapy (N, %)	14 (53.8%)	13 (52%)	0.93
Tumor localization (N, %)			0.85
Lower third	3 (11.5%)	4 (16%)	
Middle third	18 (69.2%)	18 (72%)	
Upper third	5 (19.3%)	3 (12%)	
Distance of the anastomosis from the anal verge (cm), (range), median	4 ± 1 (from 3 to 5), 4.5	4 ± 2 (from 2 to 6), 4.6	0.92

* early closure group—26 patients, standard closure group—25 patients; simple descriptive analysis was used.

**Table 2 jcm-10-00768-t002:** Low anterior resection syndrome (LARS) score for patients undergoing early (EC) or standard ileostomy closure (SC).

	EC (*N* = 26)	SC (*N* = 25)	*p* Value
Low anterior resection syndrome (LARS) score	no LARS—14 (54%) minor LARS—6 (23%) major LARS—6 (23%)	no LARS—11 (44%) minor LARS—6 (24%) major LARS—8 (32%)	0.86

Two groups were compared using Student’s *t*-test.

**Table 3 jcm-10-00768-t003:** Risk factors for worse bowel function and quality of life.

**Low Anterior Resection Syndrome (LARS)**			
	Major LARS *N* = 14	No major LARS *N* = 37	*p* *
Early closure	Yes: 8 No: 6	Yes: 18 No: 19	0.59
Complications	Yes: 2 No: 12	Yes: 4 No: 33	0.73
Gender	Male: 6 Female: 8	Male: 23 Female: 14	0.21
Neoadjuvant treatment	Yes: 8 No: 6	Yes 19 No 18	0.35
**Quality of Life**			
	Good quality of life *N* = 16	Poor quality of life *N* = 35	*p*
Early closure	Yes: 9 No: 7	Yes: 17 No: 18	0.61
Complications	Yes: 4 No: 12	Yes: 2 No: 33	0.06
Gender	Male: 9 Female: 7	Male: 20 Female: 15	0.95
Neoadjuvant treatment	Yes: 9 No: 8	Yes: 18 No: 17	0.72

* chi-squared test was used.

## Data Availability

Data available on request due to restrictions eg privacy or ethical.

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
