# Peer review of "Quality of Life and Bowel Function Following Early Closure of a Temporary Ileostomy in Patients with Rectal Cancer: A Report from a Single-Center Randomized Controlled Trial"

_jcm, 2021, doi:10.3390/jcm10040768_

Round 1
Reviewer 1 Report
The authors should say whether any patients had undergone adjuvant chemotherapy and say if this had an impact on the results.
In clinical practice, adjuvant chemotherapy is one of the main factors determining the timing of ileostomy closure.
Author Response
Dear Reviewer,
Thank you for your letter and constructive comments concerning our manuscript entitled “Quality of life and bowel function following early closure of a temporary ileostomy in patients with rectal cancer: a report from a single centre randomized controlled trial”. The paper was revised substantially. Following changes have been made. They are as follows:
Revised paragraphs, sentences, words are below:
The authors should say whether any patients had undergone adjuvant chemotherapy and say if this had an impact on the results.
In clinical practice, adjuvant chemotherapy is one of the main factors determining the timing of ileostomy closure.
Thank you for your valuable comment! Group I patients (early closure group) had no chemotherapy before the ileostomy take down as this was performed within one month of index operation (rectal resection). In group II we did the operation between the chemotherapy cycles.
Moreover, in our institution apart from the trial, most of the time we do the closure in between the cycles of chemotherapy (one week following the chemotherapy).
The explanation was also added in the Methods part.
Thank you very much indeed.
Sincerely
Reviewer 2 Report
I read your paper with much interest, and I have some questions and comments.
- I couldn't understand the meaning of a sentence in the line of 198 (`The drop of `).
-
What do you think is the reason why there was no significant difference on patients’ QOL in between two groups of EC and SC?
I thought the timing of your inquiry was not appropriate. The inquiry timing (about 3 years after ileostomy closure) would be too late. Moreover, I thought firstly you should compare patients’ score in other timings, such as a) soon after LAR, b) soon after ileostomy closure, and c) after several months from ileostomy closure. And then you had better compare their score between two groups EC and SC. Because patients have many troubles with ileostomy, and they also have other troubles without ileostomy (bowel disfunction). And the troubles would be proven in several months when they undergo ileostomy closure or they get used to the bowel condition by themselves.
Your opinion in line 209-210 is right, I think. So I would like you to have deeper discussion. Your simple comparison between EC and SC could not answer your question.
-
Do you have any trouble in patients with ileostomy, such as watery diarrhea, dehydration, or electrolyte abnormality during adjuvant chemotherapy for 3 to 6 months? These complications would affect patients QOL, so if you have some data, please consider a comparison between patients with or without adjuvant chemotherapy.
Author Response
Dear Reviewer,
Thank you for your letter and constructive comments concerning our manuscript entitled “Quality of life and bowel function following early closure of a temporary ileostomy in patients with rectal cancer: a report from a single centre randomized controlled trial”. The paper was revised substantially. Following changes have been made. They are as follows:
Revised paragraphs, sentences, words are below:
I read your paper with much interest, and I have some questions and comments.
- I couldn't understand the meaning of a sentence in the line of 198 (`The drop of `).
„The drop of“ was changed to „The patient drop of“. Thank you for your comment.
- What do you think is the reason why there was no significant difference on patients’ QOL in between two groups of EC and SC?
I thought the timing of your inquiry was not appropriate. The inquiry timing (about 3 years after ileostomy closure) would be too late. Moreover, I thought firstly you should compare patients’ score in other timings, such as a) soon after LAR, b) soon after ileostomy closure, and c) after several months from ileostomy closure. And then you had better compare their score between two groups EC and SC. Because patients have many troubles with ileostomy, and they also have other troubles without ileostomy (bowel disfunction). And the troubles would be proven in several months when they undergo ileostomy closure or they get used to the bowel condition by themselves.
We absolutely agree with your suggestion that quality of life (bowel function probably as well) should have been assessed in a longitudinal manner – at few different time spots. However, this is one of our study limitations, that could not be corrected now. Regarding the bowel function, our assessment is very important. As you know, most studies (including our own results) show that LARS is a long-term issue and it stabilizes (reaches plato) in around 1-2 years following the ileostomys closure or initial rectal resection if no stoma needed. So if we think that timing of ileostomy closure is a risk factor of worse LARS, then assessing the bowel function at least two years following the surgery makes sense. However, we do agree that adding one more time spot (for example one year following the surgery) would have increased the value of the study.
Your opinion in line 209-210 is right, I think. So I would like you to have deeper discussion. Your
simple comparison between EC and SC could not answer your question.
Still we do think that also a small sample size, we think it adds scientific evidence on timing on ileostomy closure. We added few sentence to expand the discussion.
3. Do you have any trouble in patients with ileostomy, such as watery diarrhea, dehydration, or electrolyte abnormality during adjuvant chemotherapy for 3 to 6 months? These complications would affect patients QOL, so if you have some data, please consider a comparison between patients with or without adjuvant chemotherapy.
We totally agree that the ileostomy related complications would definitely affect the quality of life negatively. However, as our sample size was small, and even smaller number underwent chemotherapy, we had no patients with the complications mentioned above. This was added to the Discussion.
Thank you very much indeed for your valuable comments. It improved the manuscript.
Sincerely
Reviewer 3 Report
I think this is a very interesting study. However, there are some concerns.
I think this is a very interesting study. However, there are some concerns.
The first is that the sample size is still small, as stated as a limiting factor.
The second is that, as also mentioned as a limitation, the assessment of sexual function and urinary function may be essential in this study.
Author Response
Dear Reviewer,
Thank you for your letter and constructive comments concerning our manuscript entitled “Quality of life and bowel function following early closure of a temporary ileostomy in patients with rectal cancer: a report from a single centre randomized controlled trial”. The paper was revised substantially. Following changes have been made. They are as follows:
Revised paragraphs, sentences, words are below:
I think this is a very interesting study. However, there are some concerns.
The first is that the sample size is still small, as stated as a limiting factor.
We totally agree with you. However, we just could not increase the number, as the study was stopped prematurely.
The second is that, as also mentioned as a limitation, the assessment of sexual function and urinary function may be essential in this study.
We agree with you. However, the goal of our study was assessment of bowel function and not the pelvic organ function. Of course, we could call all the patients and as their urogenital function. Nevertheless, the time following surgery is now around 3 years and in most patients, urogenital dysfunction is present in a short-term. In addition, it is a great idea for future studies!
Thank you very much indeed.
Sincerely
Reviewer 4 Report
The author and team had worked and produced an excellent study which will be helpful for doctors for further research and treatment on the development of prognosis.
It would be better if they implement the following points
- results can be represented in the statistical distribution of outcomes such as KM curve or at least any funnel plot diagrams. So that readers can understand easily.
- The quality of Figure 1 can be improved
- English language editing can be improved
- Table 1- Should be tabulated well. The brackets numbers are tagged properly on the heading. It's confusing whether its a percentage or n numbers?
- Same like table 2- confusing what's near to Standard deviation values? It should be properly given title box of the concerned sectio
Author Response
Dear Reviewer,
Thank you for your letter and constructive comments concerning our manuscript entitled “Quality of life and bowel function following early closure of a temporary ileostomy in patients with rectal cancer: a report from a single centre randomized controlled trial”. The paper was revised substantially. Following changes have been made. They are as follows:
Revised paragraphs, sentences, words are below:
The author and team had worked and produced an excellent study which will be helpful for doctors for further research and treatment on the development of prognosis.
It would be better if they implement the following points
1. results can be represented in the statistical distribution of outcomes such as KM curve or at least any funnel plot diagrams. So that readers can understand easily.
Table 2 and 4 changed to diagrams as per suggestion. Thank you.
2.The quality of Figure 1 can be improved
Thank you for your comment. This is the standard flow chart. And the headings of the squares are also standard. The quality was improved as suggested.
3. English language editing can be improved
The paper was revised by native English speaker.
4. Table 1- Should be tabulated well. The brackets numbers are tagged properly on the heading. It's confusing whether its a percentage or n numbers?
The table was reconstructed as per suggestion.
5. Same like table 2- confusing what's near to Standard deviation values? It should be properly given title box of the concerned section
Table 2 was changed to curve.
Thank You very much indeed. The manuscript now improved a lot.
Round 2
Reviewer 2 Report
Your paper was adequately revised according to reviewers' question, and became more interesting for readers.Author Response
Dear Reviewer,
Thank you for the comment!
Sincerely
Reviewer 3 Report
Thanks for the reply.
It is well stated as Limitation, so I think it is acceptable.
Author Response
Dear Reviewer,
Thank you for the kind comments.
Sincerely yours
This manuscript is a resubmission of an earlier submission. The following is a list of the peer review reports and author responses from that submission.
Round 1
Reviewer 1 Report
The comments of the Reviewers were sufficiently adressed.
Author Response
Dear Reviewer,
Thank you for your previous comments.
Sincerely
Audrius Dulskas
Reviewer 2 Report
In a cross-sectional study, the authors assessed quality of life and bowel function of patients who were previously enrolled in a randomised trial comparing early and standard closure of ileostomy, prematurely terminated due to the safety reasons.
My main concerns are the following:
1) I wonder what is the benefit of this study considering that the clinical data have shown the inferiority of early stoma closure, especially as the study is likely to be underpowered to detect any difference and that it is likely to be biased in the selection of participants.
2) It is unclear why the authors decided to report only QoL and bowel function after a rather long follow-up (17 to 97 month from stoma closure), as it is often the case that differences reduce overtime, and considering that in the study QoL was assessed at 3, 6, 9 and 12 moths after surgery (according to the trial registration).
3) Affirming that “complications following the ileostomy closure had no negative effect on QoL or bowel function” (p 11 line 176-177) is overstated considering the study limitations (see previous points).
Additional concerns:
I would use analyses of secondary outcome measures instead of secondary analyses, which usually refers to analyzing existing data that were collected for other purposes.
p 4 lines 86-87 The sentence “Most authors would agree …” needs clarification.
p 5 line 113 It is unclear whether the patients were contacted by phone and then sent the questionnaires by mail/email or the questionnaires were also administered by phone.
p . 10 line 165-166 It is stated “Complications did not negatively affect the bowel function (p=0,731) nor quality of life assessed as individual questions nor global score (p>0.05).” but it is unclear how this was assessed and whether participants in this study were representative of the complication rate in the clinical study.
Author Response
Reviewer #2: In a cross-sectional study, the authors assessed quality of life and bowel function of patients who were previously enrolled in a randomised trial comparing early and standard closure of ileostomy, prematurely terminated due to the safety reasons.
My main concerns are the following:
1) I wonder what is the benefit of this study considering that the clinical data have shown the inferiority of early stoma closure, especially as the study is likely to be underpowered to detect any difference and that it is likely to be biased in the selection of participants.
Yes, we do agree that this is a major limitation of our study (as it is stated in the Limitations part). However, our study is one of the largest studies conducted so far, still we think it adds a valuable information regarding the closure of ileostomy and functional outcomes.
2) It is unclear why the authors decided to report only QoL and bowel function after a rather long follow-up (17 to 97 month from stoma closure), as it is often the case that differences reduce overtime, and considering that in the study QoL was assessed at 3, 6, 9 and 12 moths after surgery (according to the trial registration).
The trial registration was registered by our fellow resident and he registered our draft of the trial. Just before the initiation of the trial the procedure was changed – LARS score was added. As we all know, LARS symptoms improve over one year and as most studies show (including our experience), it becomes more or less stable later on. For this reason, even if the assessment of bowel function was within the rather long time lapse, we believe it does not affect our results. However, the remark is very valuable and should be included in the limitations section for the readers.
3) Affirming that “complications following the ileostomy closure had no negative effect on QoL or bowel function” (p 11 line 176-177) is overstated considering the study limitations (see previous points).
Totally agree, the statement was softened.
Additional concerns:
I would use analyses of secondary outcome measures instead of secondary analyses, which usually refers to analyzing existing data that were collected for other purposes.
Great remark! We have changed as per suggestion.
p 4 lines 86-87 The sentence “Most authors would agree …” needs clarification. The sentence is just our personal experience, without any scientific background (personal opinion). The sentence was deleted.
p 5 line 113 It is unclear whether the patients were contacted by phone and then sent the questionnaires by mail/email or the questionnaires were also administered by phone. The questionnaires were send by e-mail/mail and filled together using the phone or face-to-face. Added in the text.
p . 10 line 165-166 It is stated “Complications did not negatively affect the bowel function (p=0,731) nor quality of life assessed as individual questions nor global score (p>0.05).” but it is unclear how this was assessed and whether participants in this study were representative of the complication rate in the clinical study. We agree and we added in the Limitations section that our study probably was underpowered to assess possible effect of complications to bowel function.
Thank you for your comments.
Sincerely
Audrius Dulskas, MD, PhD
Reviewer 3 Report
REVIEW of the manuscript:
„Quality of life and bowel function following early closure of a temporary ileostomy in patients with rectal cancer: secondary analysis of data from a single centre randomized controlled trial.”
- The aim of this study, presented to me for review, was to conduct a secondary analysis of our previously published study using two questionnaires - LARS score and EORTC QLQ-C30 - to investigate if early (1 month) vs late (3 months) stoma closure has a different impact on bowel function and quality of life in the late postoperative period.
- In my opinion, it is an interesting report, but it requires the following remarks to be taken into account.
- Please spell out the abbreviations first before using them:
- LARS score - Low Anterior Resection Syndrome score
- EORTC QLQ-C30 – eng. European Organisation for Research and Treatment of Cancer Quality of Life Questionaire (?)
- Line 101: If there was a locally advanced tumor, preoperative chemoradiotherapy was initiated.
The chemoradiotherapy used in some patients could have to influence the test result, but this has not been taken into account.
- What were the inclusion criteria for the study group – please specify them?
What was the result of the histopathological examination? Whether such an assessment was carried out, probably yes – it is worth emphasizing the inclusion criteria for the study.
- What were the exclusion criteria?
- Please provide the characteristics of the study group (age – mean +SD, median, presence of comorbidities (only 3 were mentioned, whether the patients had other diseases?), which may significantly interfere with the assessment.
The description of the study groups shows a gender division, please explain – was there a separate analysis for men and women?
- Line 204-5: We have not assessed the possible risk factors (height of anastomosis, preoperative radiotherapy) for worse bowel function or QoL as the sample size is quite small and this was not an aim of this study.
Measuring the quality of life is difficult, multidimensional, and laborious. I understand that this has not been the subject of the study, but all factors that affect the quality of life of patients should be analyzed. It is worth characterizing the factors that may affect the quality of life – the dose of radiation during radiotherapy.
- Did the authors take into account all medications used by patients with, for example, antidepressants, hypnotics, antidiarrheal medications, and supplements? Were the patients consulted by a clinical psychologist and dietitian?
- I propose that the authors include the used version of the questionnaire and refer to the validated version of the questionnaire.
„The LARS score was developed in Denmark, and Swedish, Spanish, and German versions have been validated. The aim of this study was to validate the English translation of the LARS score in British rectal cancer patients.”
[https://doi.org/10.1111/codi.12952
Validation of the English translation of the low anterior resection syndrome score].
- References should be supplemented with current articles. Out of 37 references, only 16 come from the last 5 years.
According to my opinion, this manuscript is suitable for publication after major revision, which will increase the value of the presented work.
Author Response
A point-by-point response to the editor's comments:
Dear Editor,
Thank you for your letter and constructive comments concerning our manuscript entitled “ Quality of life and bowel function following early closure of a temporary ileostomy in patients with rectal cancer: secondary analysis of data from a single centre randomized controlled trial”. The paper was revised substantially. Following changes have been made. They are as follows:
Reviewer#3 REVIEW of the manuscript:„Quality of life and bowel function following early closure of a temporary ileostomy in patients with rectal cancer: secondary analysis of data from a single centre randomized controlled trial.”
The aim of this study, presented to me for review, was to conduct a secondary analysis of our previously published study using two questionnaires - LARS score and EORTC QLQ-C30 - to investigate if early (1 month) vs late (3 months) stoma closure has a different impact on bowel function and quality of life in the late postoperative period.
In my opinion, it is an interesting report, but it requires the following remarks to be taken into account.
Please spell out the abbreviations first before using them:
LARS score - Low Anterior Resection Syndrome score
EORTC QLQ-C30 – eng. European Organisation for Research and Treatment of Cancer Quality of Life Questionaire (?)
The abbreviations were spelled out.
Line 101: If there was a locally advanced tumor, preoperative chemoradiotherapy was initiated. The chemoradiotherapy used in some patients could have to influence the test result, but this has not been taken into account. We have added these calculations to the Results section.
What were the inclusion criteria for the study group – please specify them? Inclusion criteria specified.
What was the result of the histopathological examination? Whether such an assessment was carried out, probably yes – it is worth emphasizing the inclusion criteria for the study. The results of histopathological examination inserted.
What were the exclusion criteria? Exclusion criteria inserted.
Please provide the characteristics of the study group (age – mean +SD, median, presence of comorbidities (only 3 were mentioned, whether the patients had other diseases?), which may significantly interfere with the assessment. No other comorbidities were present. Other data was updated and inserted.
The description of the study groups shows a gender division, please explain – was there a separate analysis for men and women? No, we did not performed the separate analysis.
Line 204-5: We have not assessed the possible risk factors (height of anastomosis, preoperative radiotherapy) for worse bowel function or QoL as the sample size is quite small and this was not an aim of this study. Measuring the quality of life is difficult, multidimensional, and laborious. I understand that this has not been the subject of the study, but all factors that affect the quality of life of patients should be analyzed. It is worth characterizing the factors that may affect the quality of life – the dose of radiation during radiotherapy. We have inserted the calculations to the Results section.
Did the authors take into account all medications used by patients with, for example, antidepressants, hypnotics, antidiarrheal medications, and supplements? Were the patients consulted by a clinical psychologist and dietitian? The patients were asked about the medications they used by phone – none of them have been any of mentioned drugs constantly. In our country we d not have a specific care for patients with LARS. Moreover, even having the major LARS, patients refuse for any treatment or even the dietary changes.
I propose that the authors include the used version of the questionnaire and refer to the validated version of the questionnaire. „The LARS score was developed in Denmark, and Swedish, Spanish, and German versions have been validated. The aim of this study was to validate the English translation of the LARS score in British rectal cancer patients.” [https://doi.org/10.1111/codi.12952 Validation of the English translation of the low anterior resection syndrome score]. We have included the reference as per suggestion.
References should be supplemented with current articles. Out of 37 references, only 16 come from the last 5 years. Yes, we agree that most of the references are rather old. However, quality of life following the ileostomy take-down is underreported in more recent literature.
According to my opinion, this manuscript is suitable for publication after major revision, which will increase the value of the presented work.
Thank you for your comments.
Sincerely
Audrius Dulskas, MD, PhD
Round 2
Reviewer 2 Report
While the authors have improved the manuscript, the main issues remain unaddressed.
Specifically, it is unclear what would be the benefit of knowing that there are or there are not differences in QoL and LARS if the clinical data were so clear that the study was prematurely stopped.
Having a bigger sample size than previous studies is not a sufficient reason to publish the study if it is likely to be underpowered to detect differences and, most importantly, biased in the selection of participants (i.e., those who suffered most severe complications).
Finally, even if the LARS score was added after the trial registration, it is unclear why this changed the timing of assessment of QoL and bowel function that was registered. Additionally, as the authors agree that bowel function improves overtime, the very long and varied follow-up included in the sample further reduces the possibility to ascertain any difference between the groups.
Author Response
Dear Reviewer,
Thank you for your constructive and valuable comments. You appreciate your in-depth analysis of our work.
1)Specifically, it is unclear what would be the benefit of knowing that there are or there are not differences in QoL and LARS if the clinical data were so clear that the study was prematurely stopped.
The bowel function and quality of life should be also considered when we think (or when we are deciding) when to take down the ileostomy. Some studies do show that living with stoma negatively affects the quality of life [1-3]. Moreover, some studies on LARS show worse bowel function in patients living longer with defunctioning ileostomy [4]. Our finding are important as they show that in a short-term (up to three months) there was no difference in QoL or bowel function.
- Dabirian A, Yaghmaei F, Rassouli M, Tafreshi MZ. Quality of life in ostomy patients: a qualitative study. Patient Prefer Adherence. 2010 Dec 21;5:1-5. doi: 10.2147/PPA.S14508. PMID: 21311696; PMCID: PMC3034300.
- Silva JS, et al. Quality of Life (QoL) Among Ostomized Patients – a cross-sectional study using Stoma-care QoL questionnaire about the influence of some clinical and demographic data on patients’ QoL. J Coloproctol 2019;39:48-55
- Scarpa M, Barollo M, Polese L, Keighley MR. Quality of life in patients with an ileostomy. Minerva Chir. 2004;59(1):23-9.
- Hughes DL, Cornish J, Morris C; LARRIS Trial Management Group. Functional outcome following rectal surgery-predisposing factors for low anterior resection syndrome. Int J Colorectal Dis. 2017 May;32(5):691-697.
2) Having a bigger sample size than previous studies is not a sufficient reason to publish the study if it is likely to be underpowered to detect differences and, most importantly, biased in the selection of participants (i.e., those who suffered most severe complications).
We did not select only the patients with most severe complications. We have included all the patients who underwent rectal resection with ileostomy and without radiologically evident leak. Moreover, these are not the patients with most severe complications (23% (6 of 26) in EC vs 12% (3 of 25).
3) Finally, even if the LARS score was added after the trial registration, it is unclear why this changed the timing of assessment of QoL and bowel function that was registered. Additionally, as the authors agree that bowel function improves overtime, the very long and varied follow-up included in the sample further reduces the possibility to ascertain any difference between the groups.
As we explained previously, the primary study protocol was registered by our fellow. We have changed the protocol, without changing it in Trials.gov – this is our mistake. The reason of changes was that 3 months periods are too short for bowel function assessment and our goal was to assess the bowel function when it is rather stable. A longitudinal assessment would have been preferable. However, the minimum follow‐up for all participants was almost 2 years (the average of 36 months) after restoration of bowel continuity and by this stage functional outcomes would be expected to be relatively stable [1, 2].
- Pieniowski EHA, Palmer GJ, Juul T, Lagergren P, Johar A, Emmertsen KJ, Nordenvall C, Abraham-Nordling M. Low Anterior Resection Syndrome and Quality of Life After Sphincter-Sparing Rectal Cancer Surgery: A Long-term Longitudinal Follow-up. Dis Colon Rectum. 2019 Jan;62(1):14-20.
- Sandberg S, Asplund D, Bisgaard T, Bock D, González E, Karlsson L, Matthiessen P, Ohlsson B, Park J, Rosenberg J, Skullman S, Sörensson M, Angenete E. Low anterior resection syndrome in a Scandinavian population of patients with rectal cancer: a longitudinal follow-up within the QoLiRECT study. Colorectal Dis. 2020 Oct;22(10):1367-1378.
Thank You in advance
Sincerely
Audrius Dulskas, MD, PhD
Reviewer 3 Report
I would like to thank you authors for quick and adequate manuscript correction, no more comment
I would like to thank you, authors, for the quick and adequate manuscript correction, no more comments.
Author Response
Dear Reviewer,
Thank you for valuable comments.
Sincerely
Audrius Dulskas, MD, PhD